# Slow oscillation-spindle coupling predicts enhanced memory formation from childhood to adolescence

**Michael A Hahn[1,2]\*, Dominik Heib[1,2], Manuel Schabus[1,2], Kerstin Hoedlmoser[1,2]†\*, Randolph F Helfrich[3]†\***

[1]Department of Psychology, Laboratory for Sleep, Cognition and Consciousness Research, University of Salzburg, Salzburg, Austria; [2]Centre for Cognitive Neuroscience Salzburg (CCNS), University of Salzburg, Salzburg, Austria; [3]Hertie-Institute for Clinical Brain Research, University of Tübingen, Tübingen, Germany

**Abstract** Precise temporal coordination of slow oscillations (SO) and sleep spindles is a fundamental mechanism of sleep-dependent memory consolidation. SO and spindle morphology changes considerably throughout development. Critically, it remains unknown how the precise temporal coordination of these two sleep oscillations develops during brain maturation and whether their synchronization indexes the development of memory networks. Here, we use a longitudinal study design spanning from childhood to adolescence, where participants underwent polysomnography and performed a declarative word-pair learning task. Performance on the memory task was better during adolescence. After disentangling oscillatory components from 1/f activity, we found frequency shifts within SO and spindle frequency bands. Consequently, we devised an individualized cross-frequency coupling approach, which demonstrates that SO-spindle coupling strength increases during maturation. Critically, this increase indicated enhanced memory formation from childhood to adolescence. Our results provide evidence that improved coordination between SOs and spindles indexes the development of sleep-dependent memory networks.

**\*For correspondence:**
michael.hahn@sbg.ac.at (MAH);
kerstin.hoedlmoser@sbg.ac.at
(KH);
randolph.helfrich@gmail.com
(RFH)

†These authors contributed
equally to this work

**Competing interests:** The
authors declare that no
competing interests exist.

**Reviewing editor:** Saskia
Haegens, Columbia University
College of Physicians and
Surgeons, United States

## Introduction

Active system memory consolidation theory proposes that sleep-dependent memory consolidation is orchestrated by three cardinal sleep oscillations (*Diekelmann and Born, 2010*; *Helfrich et al., 2019*; *Klinzing et al., 2019*; *Mölle et al., 2011*; *Piantoni et al., 2013*; *Rasch and Born, 2013*; *Staresina et al., 2015*): (1) Hippocampal sharp-wave ripples represent the neuronal substrate of memory reactivation (*Vaz et al., 2019*; *Wilson and McNaughton, 1994*; *Zhang et al., 2018*), (2) thalamo-cortical sleep spindles are thought to promote long-term potentiation (*Antony et al., 2018*; *De Gennaro and Ferrara, 2003*; *Rosanova and Ulrich, 2005*; *Schönauer, 2018*; *Schönauer and Pöhlchen, 2018*), while (3) neocortical SOs provide temporal reference frames where memory can be replayed, potentiated and eventually transferred from the short-term storage in the hippocampus to the long-term storage in the neocortex, rendering memories increasingly more stable (*Chauvette et al., 2012*; *Diekelmann and Born, 2010*; *Frankland and Bontempi, 2005*; *Rasch and Born, 2013*). Importantly, these three oscillations form a temporal hierarchy, where ripples and spindles are nested in SO peaks, with ripples also being locked to spindle troughs. This hierarchy likely constitutes an endogenous timing mechanism to ensure that the neocortical system is in an optimal state to consolidate new hippocampus-dependent memories (*Chauvette et al., 2012*; *Clemens et al., 2011*; *Helfrich et al., 2019*; *Klinzing et al., 2016*; *Klinzing et al., 2019*; *Latchoumane et al., 2017*; *Niethard et al., 2018*; *Piantoni et al., 2013*; *Staresina et al., 2015*).

**eLife digest** Sleep is essential for consolidating the memories that we made during the day. As we lie asleep, unconscious, our brain is busy processing the day's memories, which travel through three parts of the brain before they are filed away. First, the hippocampus, the part of the brain that stores memories temporarily, replays the memories of the day. Then the reactivated memories pass through the thalamus, a central crossroads in the brain, so they can be embedded in the neocortex for long-term storage.

Neuroscientists can eavesdrop on the brain at work, day or night, using a technique called EEG. Short for electroencephalogram, an EEG detects brain waves like the bursts of electrical activity known as sleep spindles and slower sleep waves called slow oscillations. These two brain wave patterns represent how the brain processes memories as people sleep – and it is all about timing. If the two patterns are running in sync, then the brain's memory systems are thought to be communicating well and memories are more likely to be stored.

But patterns of sleep spindles and slow oscillations change dramatically between childhood and adolescence. Memory consolidation also improves in those formative years. Still, it is not yet known if better synchronization between sleep spindles and slow oscillations explains how memory formation improves during this period; that is the going theory.

To test it out, Hahn et al. completed a unique study examining how well a group of 33 children could store memories, and then again when the same group were teenagers. Both times, the group was asked to memorise and then recall a set of words before and after a full night's sleep. Hahn et al. measured how much their memory recall improved and whether their brain wave patterns were in sync, looking for any changes between childhood and adolescence. This showed that children whose sleep spindles stacked better with their slow oscillations had improved memory formation once they became teenagers.

This work highlights how communication between memory systems in the brain improves as children age, and so does memory. Moreover, it suggests that if disturbances were to be detected in patterns of sleep spindles and slow oscillations, there might be some problem with memory storage. It also points to brain stimulation as a possible treatment option for such problems in the future.

---

Recent findings indicate that the precise temporal coordination of SO-spindle coupling is deteriorating over the lifespan, which contributes to age-related memory decline (*Helfrich et al., 2018b*; *Muehlroth et al., 2019*; *Winer et al., 2019*). It is currently unclear if similar principles apply to brain maturation and how the dynamic interplay of SOs and spindles is initiated. Critically, the transition from childhood to adolescence is marked by considerable changes in sleep architecture and cognitive abilities similar to the transition from young adulthood to old age (*Carskadon et al., 2004*; *Huber and Born, 2014*; *Iglowstein et al., 2003*; *Ohayon et al., 2004*; *Shaw, 2007*; *Shaw et al., 2006*). Previous research mainly focused on the individual development of SOs and sleep spindles across brain maturation, showing that these cardinal sleep oscillations undergo a substantial evolution in their defining features such as amplitude, frequency, distribution and occurrence (*Campbell and Feinberg, 2009*; *Campbell and Feinberg, 2016*; *Goldstone et al., 2019*; *Hahn et al., 2019*; *Kurth et al., 2010*; *Nicolas et al., 2001*; *Purcell et al., 2017*; *Shinomiya et al., 1999*; *Tarokh and Carskadon, 2010*).

Currently, two major obstacles hamper our understanding of how the precise temporal interplay between SOs and spindles predicts brain development and memory formation. First, pronounced changes in sleep oscillatory activity pose major methodological challenges for assessing and comparing SOs and sleep spindles across the age spectrum (*Muehlroth and Werkle-Bergner, 2020*). Second, memory performance was rarely tested in developmental sleep studies, thus, impeding our understanding of the functional significance of temporal SO-spindle coupling for memory formation.

Here, we leverage a unique longitudinal study design from childhood to adolescence to investigate how SO-spindle coupling emerges during development and infer its functional significance for developing memory networks. To account for the substantial morphological alterations of SO and spindle morphology across brain maturation, we developed a principled methodological approach

to assess SO-spindle coupling. We utilized individualized cross-frequency coupling analyses, which enable a clear demonstration of SO-sleep spindles coupling during both developmental stages. Critically, over the course of brain maturation from childhood to adolescence, more spindles are tightly coupled to SOs, which directly predicts improved memory formation.

## Results

Using a longitudinal study design, we tested 33 healthy participants during childhood (age: 9.5 ± 0.8 years; mean ± SD) and during adolescence (age: 16 ± 0.9 years). At both time points participants underwent full-night ambulatory polysomnography at their home during two nights (adaptation and experimental night; *Figure 1A*) and performed a declarative memory task during the experimental night (*Figure 1B*). After encoding, participants recalled word pairs before and after a full night of sleep. As previously shown (*Hahn et al., 2019*), memory recall improved from childhood to adolescence (*Figure 1C*; $F_{1,32}$ = 38.071, p<0.001, $\eta^2$ = 0.54) and immediate recall was better than delayed recall ($F_{1,32}$ = 6.408, p=0.016, $\eta^2$ = 0.17; Maturation*Recall Time interaction: $F_{1,32}$ = 2.059, p=0.161, $\eta^2$ = 0.06). Next, we assessed the relationship of sleep-dependent memory consolidation (delayed recall – immediate recall) between childhood and adolescence and found no correlation between the two maturational stages (*Figure 1—figure supplement 1A*; for a direct comparison of sleep-dependent memory consolidation see *Figure 1—figure supplement 1B*). During adolescence, memory consolidation was superior after a sleep retention interval compared to a wake retention interval (*Figure 1—figure supplement 1C*), indicating a beneficial effect of sleep on memory.

### Oscillatory signatures of NREM sleep during childhood and adolescence

To investigate whether SO-spindle coupling accounts for enhanced memory formation from childhood to adolescence, we first assessed the oscillatory signatures of NREM (2 and 3) sleep. We compared spectral estimates during childhood and adolescence using cluster-based permutation tests (*Maris and Oostenveld, 2007*) across frequencies from 0.1 to 20 Hz (*Figure 2A*; at electrode Cz). We found that EEG power significantly decreased from childhood to adolescence between 0.1 to 13.6 Hz (cluster test: p<0.001, d = −2.74) and 14.6 to 20 Hz (cluster test: p<0.001, d = −1.60; *Figure 2A*). However, inspection of the underlying spectra revealed that this effect was driven by (I) an overall offset of the 1/f component of the power spectrum on the y-axis and (II) by a shift of the peak frequency in the spindle band. In order to mitigate the prominent power difference, we first z-normalized the signal in the time domain, which alleviated the differences above ~15 Hz (*Figure 2B*). This analysis showed increased spectral power during childhood from 0.3 to 8.4 Hz (cluster test: p=0.002, d = −084), which was broadband and not oscillatory in nature. In addition power differences between 10.6 to 12.8 Hz (cluster test: p=0.040, d = −1.07) and 13.4 and 14.8 Hz (cluster test: p=0.046, d = 1.12) directly reflected the spindle peak frequency shift from childhood to adolescence. To account for the differences in broadband 1/f and oscillatory components, we disentangled the 1/f fractal component (*Figure 2C*) from the oscillatory residual (*Figure 2D*) by means of irregular-resampling auto-spectral analysis (IRASA *Helfrich et al., 2018b*; *Wen and Liu, 2016*). We found that a significant decrease in the fractal component between 0.3 and 10.8 Hz from childhood to adolescence (*Figure 2C*; cluster test: p<0.013, d = −0.90), accounted for the prominent broadband power differences as observed in *Figure 2A*. To assess true oscillatory brain activity, we subtracted the fractal component (*Figure 2C*) from the normalized power spectrum (*Figure 2B*), to isolate SO and spindle oscillations in the frequency domain (*Figure 2D*).

Based on the oscillatory residuals, we then extracted the individual peak frequency and the corresponding amplitude in the SO and sleep spindle range for each electrode in every participant during childhood and adolescence. After discounting 1/f effects, we found that spindle amplitude (*Figure 2E*) increased in a centro-parietal cluster (cluster test: p=0.005, d = 0.63), whereas spindle peak frequency (*Figure 2F*) accelerated at all channels from childhood to adolescence (cluster test: p<0.001, d = 1.57). SO amplitude and frequency decreased from childhood to adolescence (*Figure 2—figure supplement 1A,B*). Both, SO and spindle features have been previously related to memory formation (*Gais et al., 2002*; *Huber et al., 2004*; *Lustenberger et al., 2017*; *Schabus et al., 2004*; *Schabus et al., 2006*). However, neither spindle nor SO amplitude or peak frequency changes explained the behavioral differences (*Figure 2—figure supplement 1C,D*). Note,

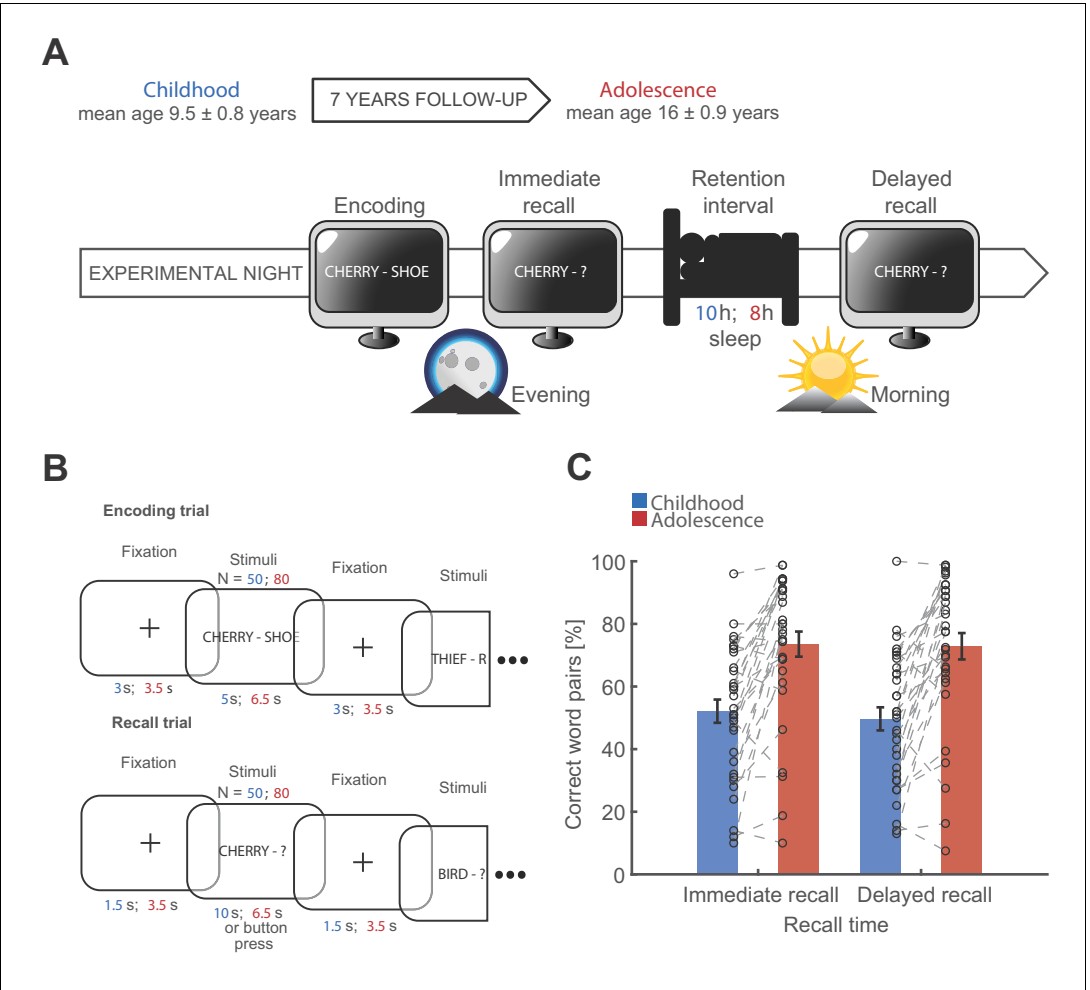

**Figure 1.** Study design and behavioral results. (**A**) Longitudinal study design. Participants were recorded during childhood (blue) and adolescence (red). Recording periods were separated by 7 years. Participants underwent two full-night ambulatory polysomnographies in their habitual sleep environment at both time points respectively. The first night served adaptation purposes. At the following experimental night, participants performed a declarative word pair learning task during which they encoded and recalled semantically non-associated word pairs before sleep. The post-sleep recall was separated by a 10 hr (childhood) and 8 hr sleep during the retention interval (adolescence). (**B**) Word pair task design. Participants encoded 50 word pairs during childhood (blue) and 80 word pairs during adolescence (red). Every word pair presentation was followed by a fixation cross. Participants were instructed to imagine a visual connection between the two words. Timing parameters are indicated in the respective colors for childhood (blue) and adolescence (red). During the recall trial, only the first word of the word pair was presented and participants had to recall the corresponding word. Participants received no performance feedback. (**C**) Behavioral results for the word pair task. Performance was measured as percentage of correctly recalled word pairs. Participants showed a higher performance during adolescence. Black circles indicate individual recall scores.

The online version of this article includes the following figure supplement(s) for figure 1:

**Figure supplement 1.** Additional behavioral analyses.

we also observed a peak in the theta band, which was unrelated to behavior (for theta peak frequency and amplitude correlations with behavior see *Figure 2—figure supplement 1E*).

## Individual features of discrete SO and sleep spindle events

After having established the cardinal features of SO and spindle oscillations during childhood and adolescence, we then individually adjusted previously used SO and spindle detection algorithms (*Helfrich et al., 2018b*; *Mölle et al., 2011*; *Staresina et al., 2015*) according to the individual peak

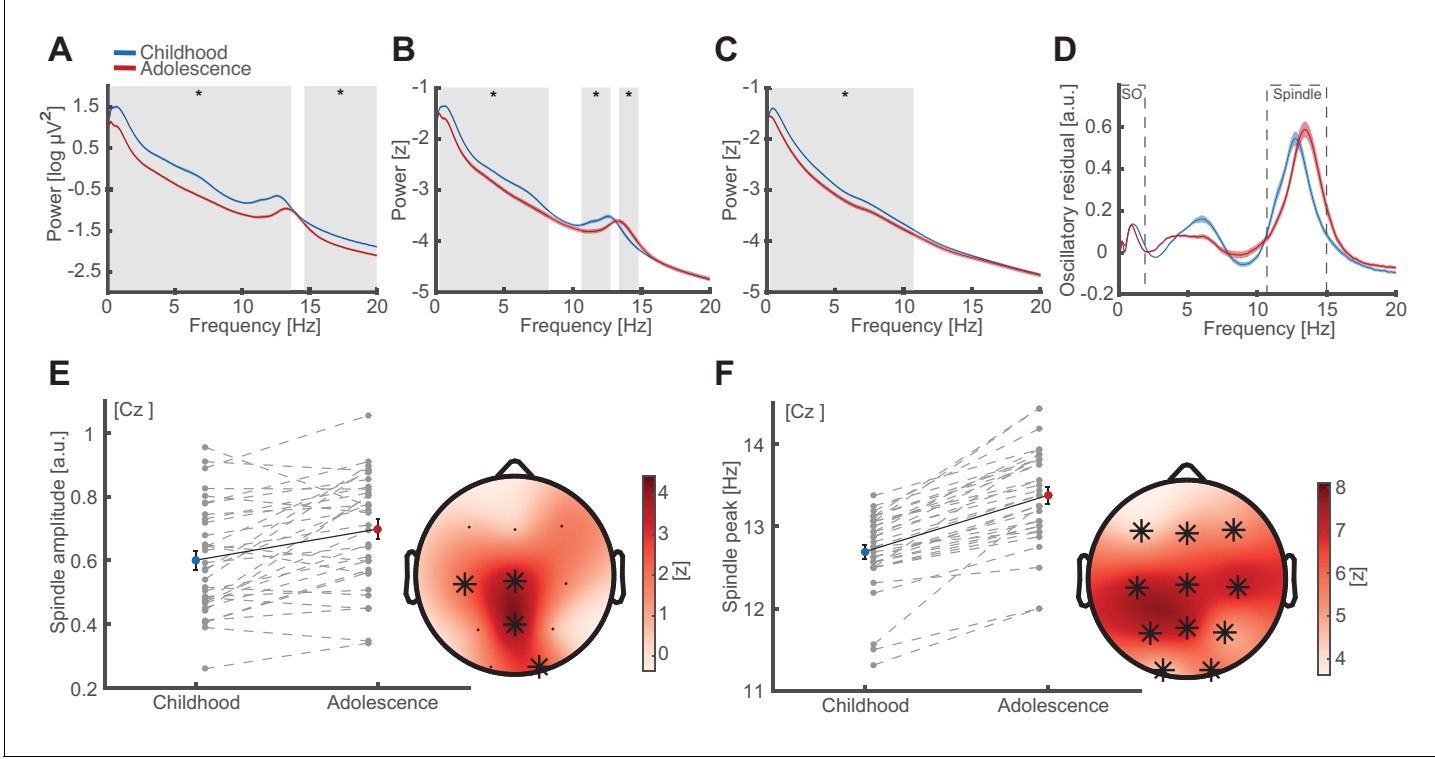

**Figure 2.** Oscillatory signatures of NREM sleep. (A) Uncorrected EEG power spectra (mean ± standard error of the mean [SEM]) during NREM (NREM2 and NREM3) sleep at Cz during childhood (blue) and adolescence (red). Grey overlays indicate significant differences (cluster-corrected). Note the overall power decrease from childhood to adolescence. (B) Z-normalized EEG power spectra. Same conventions as in (A). Significant differences indicate a change in the fractal component of power spectra (0.3–8.4 Hz) and a spindle frequency peak shift (10.6–14.8 Hz) from childhood to adolescence. (C) Extracted 1/f fractal component. Same conventions as in (A). Decrease of the fractal component (0.3–10.8 Hz) from childhood to adolescence. (D) Oscillatory residual of the NREM power spectra obtained by subtracting the fractal component (C) from the z-normalized power spectrum (B). Oscillatory residual shows clear dissociable peaks in the SO and sleep spindle frequency range (dashed boxes) during both time points, indicating true oscillatory activity. (E) Spindle amplitude development. Spindle amplitude (exemplary depiction at Cz, left, mean ± SEM) as extracted from the oscillatory residuals (D) indicating an increase in 1/f corrected amplitude within a centro-partial cluster (right) from childhood to adolescence. Grey dots represent individual values. Asterisks denote cluster-corrected two-sided p<0.05. T-scores are transformed to z-scores to indicate the difference between childhood and adolescence. (F) Spindle frequency peak development. Spindle frequency peak (mean ± SEM) as extracted from the oscillatory residual (D). Same conventions as in (E). Spindle peak frequency increases at all electrodes from childhood to adolescence.

The online version of this article includes the following figure supplement(s) for figure 2:

**Figure supplement 1.** SO feature development and correlations between oscillatory features and behavior.

**Figure supplement 2.** Individual oscillatory residuals and spindle frequency gradient.

frequencies (*Bódizs et al., 2009*; *Ujma et al., 2015*). We considered the possibility that two distinct spindle frequency peaks exist (*Anderer et al., 2001*; *Werth et al., 1997*), but inspecting the oscillatory residuals did not indicate two clearly discernable peaks in individual electrodes of the majority of participants (for exemplary oscillatory residuals see *Figure 2—figure supplement 2A*). Because we observed the typical antero-posterior spindle frequency gradient (*Cox et al., 2017*; *De Gennaro and Ferrara, 2003*; *Zeitlhofer et al., 1997*) with slower frontal and faster posterior spindles (*Figure 2—figure supplement 2B*), we used the highest peak in the spindle range at every electrode as the most representative oscillatory event for the detection algorithm. Importantly, individualized SO and spindle event detections closely followed spectral sleep patterns during childhood and adolescence (*Figure 3A,B*; event detections are superimposed in white).

Next, we quantified how many separate SO and spindle event detections co-occurred within a 2.5 s time window (reflecting ±2 SO cycles around the spindle peak; *Helfrich et al., 2019*). Note that the co-occurrence rate does not actually indicate coupled SO-spindle events but directly reflects the percentage of detected spindle events that are concomitant with detected SO events. Co-occurrence rate was higher in NREM3 than NREM2 sleep during childhood and adolescence (*Figure 3C*;

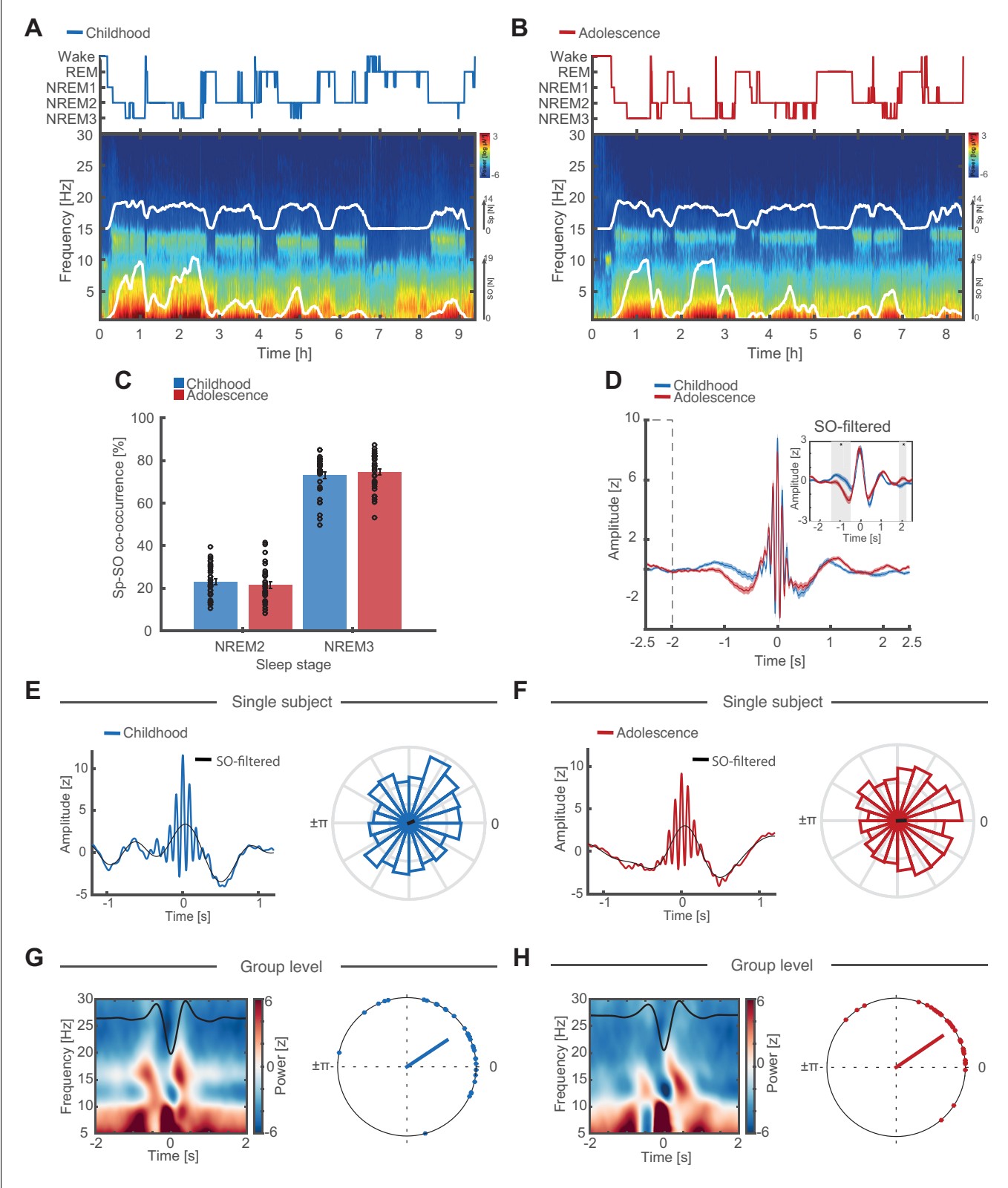

**Figure 3.** Individual features of discrete SO and sleep spindle events. (**A**) Hypnogram (top) and full-night spectrogram (bottom) at electrode Cz of an exemplary subject during childhood. White lines in the spectrogram indicate the amount of events detected by the individually adjusted detection algorithms for sleep spindles (upper trace) and SO (lower trace). (**B**) Same conventions as in (**A**) but for the same individual during adolescence. (**C**) Spindle-SO co-occurrence expressed as the percentage of SO detections that coincide ±2.5 s with a sleep spindle detection at electrode Fz during

*Figure 3 continued on next page*

*Figure 3 continued*

NREM2 and NREM3 sleep during childhood (blue) and adolescence (red). Note the high co-occurrence of spindles and SOs during NREM3 at both recording time points. (D) Grand average of z-normalized sleep spindle events (mean ± SEM) during childhood (blue) and adolescence (red) at electrode Fz with the corresponding SO-low-pass filtered (<2 Hz) EEG-trace (inset). Note that there is no baseline difference between −2.5 s and −2 s (dashed box). The significant difference in the −1.5 to -0.5 s interval (grey shaded area, SO-filtered inset) indicates an increased amount of coupled SO-sleep spindle events during adolescence. Further note, no amplitude differences in the SO-filtered signal around the spindle peak at 0 s (i.e. time point of the phase readout). Grand average spindle frequency is distorted by the individually adjusted event detection criteria. (E) SO-spindle coupling features. Data are shown for electrode Fz during NREM3. Left: Exemplary spindle-locked average for a single subject during childhood with the corresponding SO-filtered signal in black. Note that the spindle amplitude peak coincides with the maximum peak in the SO-component. Right: Normalized phase histograms of spindle events relative to SO-phase of an exemplary subject during childhood. 0˚ denotes the positive peak whereas ±π denotes the negative peak of the SO. (F) Same conventions as in (E). Left: Exemplary spindle-locked average of the same single subject as in (E) during adolescence. Note the clearer outline of a SO-component compared to during childhood indicating a stronger SO-spindle coupling. Right: Normalized phase histograms of spindle events relative to SO-phase of same exemplary subject as in (E) during adolescence. Note the reduced spread in SO-phase. (G) Left: Grand average baseline-corrected (−2 to −1.5 s) SO-trough-locked time frequency representation (TFR). Schematic SO-component is superimposed in black. Note the alternating pattern within the spindle frequency range indicating a modulation of spindle activity by SO-phase. Right: Circular plot of preferred phase (SO phase at spindle amplitude maximum) per subject during childhood. Dots indicate the preferred phase per subject. The line direction shows the grand average preferred direction. The line length denotes the mean resultant vector (i.e. sample variance of preferred phase and therefore does not represent coupling strength). Note that most subjects show spindles coupled to or just after the positive SO-peak at 0˚. Data are shown for electrode Fz during NREM3. (H) Same conventions as in (G). Left: SO-trough-locked TFR indicating a modulation in spindle activity depending on SO-phase. Right: Circular plot of preferred phase per subject during adolescence. Note that there are no preferred phase changes but an overall reduced spread in preferred phase on the group level during adolescence as indicated by a longer mean resultant vector (red line).

The online version of this article includes the following figure supplement(s) for figure 3:

**Figure supplement 1.** Uncorrected (non z-normalized) spindle-locked grand average during NREM3 at Cz for childhood (blue) and adolescence (red) with corresponding SO-filtered (<2 Hz) EEG-trace.

$F_{1,32}$ = 2334.19, p<0.001, $\eta^2$ = 0.99). Subsequently we restricted our analyses to NREM3 sleep to avoid spurious cross-frequency coupling estimates caused by the lack of simultaneous detections during NREM2 sleep (*Aru et al., 2015*; for circular plots including NREM2 sleep see *Figure 4—figure supplement 1D*).

To ensure reliable coupling estimates, we further Z-normalized individual spindle-locked data epochs in the time domain (*Figure 3D*) for all subsequent analyses to avoid possible confounding amplitude differences (*Aru et al., 2015*; *Cole and Voytek, 2017*; *Helfrich et al., 2018b*). Differences in the grand average spindle time-lock directly reflect the enhanced SO-spindle coupling, which becomes visible in the time domain when more events are precisely coupled to the SO 'up-state' (positive SO-peak). This effect can also be appreciated in single subject spindle-locked data (*Figure 3E,F*, left). Note that we found no differences in the underlying SO-component around the spindle peak (0 s, time point of phase readout), thus, confirming that the Z-normalization alleviated possible amplitude differences (*Figure 3D*, inset; for non-normalized spindle-locked data see *Figure 3—figure supplement 1*).

To further elucidate the interaction between SO phase and spindle activity, we also assessed this effect in the time-frequency domain by calculating SO-trough-locked time-frequency representations (*Figure 3G,H*, left). The alternating pattern (i.e. spindle power decreases during the 'down-state' and increases during the 'up-state') within the spindle frequency range during childhood and adolescence indicated an influence of SO phase on spindle activity.

To quantify the interplay of SO and spindle oscillations, we employed event-locked cross-frequency analyses (*Dvorak and Fenton, 2014*; *Helfrich et al., 2018b*; *Staresina et al., 2015*). While this method is mainly equivalent to other frequently used methods to assess cross-frequency coupling (*Helfrich et al., 2018a*), it can be similarly impacted by their pitfalls (*Aru et al., 2015*). Therefore, we adopted a conservative approach by first alleviating power differences (*Figure 2B* and *Figure 3D*) and establishing the presence of oscillations in the signal (*Figure 2D* and *Figure 3C*). Next, we extracted the instantaneous SO phase during every spindle peak at every electrode and for every subject. Then we calculated the preferred phase (circular mean direction) and coupling strength (phase-locking value, plv) separately during childhood and adolescence for all events at a given electrode (see *Figure 3E,F* right for exemplary phase histograms). We confirmed that both

markers of SO-spindle coupling were not confounded by differences in the total number of detected events using a bootstrapping procedure (*Figure 4—figure supplement 1C*).

## Brain maturation impacts SO-spindle coupling

First, we assessed in which phase of the SO spindles preferably occur and how maturation affects the preferred coupling direction. We found that coupling direction did not change at frontal electrodes (*Figure 3G,H*, right), where spindles were locked to the SO peak during childhood (33.5°±44.8°; circular mean ± SD) and adolescence (34.2°±35.8°; circular mean ± SD). However, we detected differences in centro-posterior clusters (central: p<0.001, d = −0.84, parieto-occipital: p<0.001, d = 0.74; *Figure 4—figure supplement 1A*), which did not correlate with behavior (*Figure 4—figure supplement 1B*).

After showing that sleep spindles are preferably locked to the SO peaks, we subsequently quantified how precisely spindles are embedded in the preferred SO phase by assessing the respective coupling strength (1 − circular variance). Coupling strength increased from childhood to adolescence across all electrodes except P4 (cluster test: p<0.001, d = 0.74; *Figure 4A*), indicating that frontal sleep spindles become more tightly locked to their respective preferred SO phase as the phase on frontal sensors remained stable across maturation. To further illustrate the impact of the coupling strength increase on spindle-SO-events, we calculated the percentage of spindle events that peaked at the preferred phase (±22.5°) as compared to all detected events (*Figure 4B*, left). The resulting metric is directly related to the coupling strength and was solely computed to further illustrate and highlight the observed effects. Concurrent with our coupling strength analyses, percentage of spindles within the preferred phase bin increased in a fronto-parietal cluster (p<0.001 d=0.74; *Figure 4B*, right) however, decreased in an occipital cluster from childhood to adolescence (p=0.007, d = 0.78). These results reveal an overall increase in SO-spindle coupling precision from childhood to adolescence.

## SO-spindle coupling development predicts memory formation

After having established SO-spindle coupling properties and demonstrating their qualitative enhancement from childhood to adolescence, we next tested the hypothesis that these changes also predict maturational differences in recall performance and sleep-dependent memory consolidation.

We utilized cluster-based correlation analyses to relate differences in coupling strength to differences in recall performance (delayed recall_adolescence – delayed recall_childhood). We observed a significant frontal cluster (p=0.0050, mean rho = 0.48; *Figure 4C*; left) showing that a stronger increase in SO-spindle coupling strength from childhood to adolescence related to improved recall performance from childhood to adolescence. This relationship was most pronounced at electrode F3 (rho = 0.57, p<0.001; *Figure 4C*; right). Notably using a non-individualized approach with a fixed frequency band (11–15 Hz) attenuated the test statistic (rho = 0.50, p=0.035, cluster-corrected; *Figure 4—figure supplement 1E*).

Next, we only considered spindle events that co-occur with detected SO-events to (1) ensure more precise coupling metrics by guaranteeing the presence of oscillations and (2) to follow the notion that the temporal proximity of these events is crucial for sleep-dependent memory consolidation (*Diekelmann and Born, 2010*; *Helfrich et al., 2018b*; *Muehlroth et al., 2019*; *Rasch and Born, 2013*). Participants with a stronger coupling strength increase also showed enhanced sleep-dependent memory consolidation (delayed recall - immediate recall) from childhood to adolescence (rho = 0.54, p=0.0011; *Figure 4D*, electrode F3).

To validate whether the SO-spindle coupling strength predicts sleep specific memory formation, we correlated coupling strength during adolescence with the difference in memory consolidation between the sleep and wake condition ([(delayed recall_sleep –immediate recall_sleep)- (delayed recall_wake –immediate recall_wake)]). However, we could not find evidence for this relationship (*Figure 4—figure supplement 1F*).

Previously, we and others reported that spindle density correlates with memory formation (*Gais et al., 2002*; *Hahn et al., 2019*). In order to test whether spindle densities potentially confound metrics of cross-frequency coupling, we first correlated the two but found no significant correlation (rho_childhood = 0.08, p=0.664; rho_adolescence = −0.115, p=0.400; rho_adolescence-childhood = −0.02, p=0.894). Next, we partialed out the influence of spindle density on the original coupling strength-

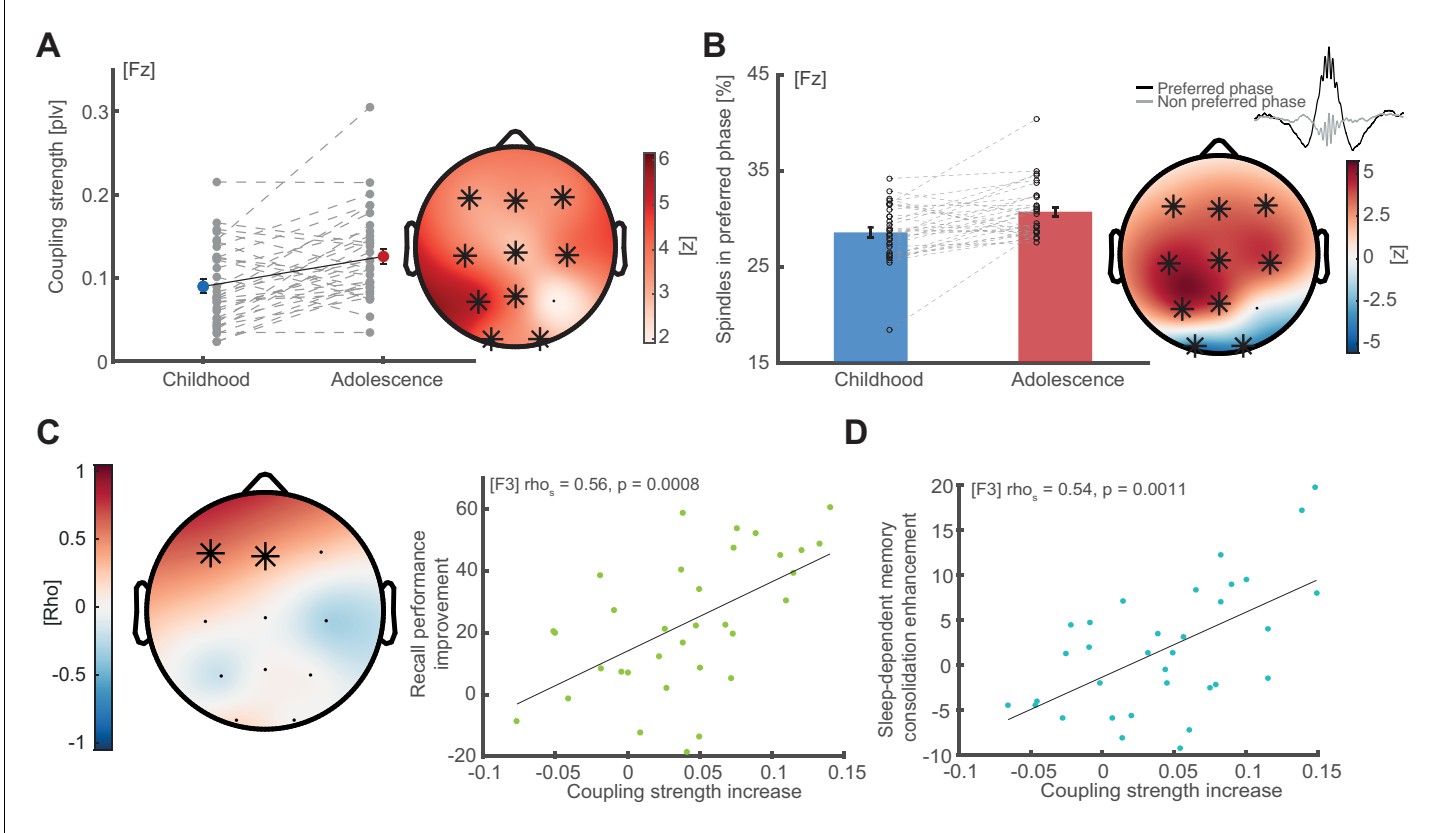

**Figure 4.** Coupling strength development and correlations with memory formation. (**A**) Coupling strength development. Coupling strength (phase locking value; mean ± SEM) increases from childhood to adolescence (exemplary data at Fz, left, grey dots indicate individual values) at all electrodes except P4 (topographical plot, right), indicating that more spindles arrive within the preferred phase during adolescence than during childhood. Asterisks indicate cluster-corrected two-sided p<0.05. T-scores are transformed to z-scores to indicate the difference between childhood and adolescence. (**B**) Spindles in preferred phase. Same conventions as in (**A**). Bar plot depicts the percentage of sleep spindles (mean ± SEM) that arrive in a ± 22.5° radius around the individual preferred phase. Like coupling strength, spindles in preferred phase increase from childhood to adolescence in a fronto-parietal cluster but decrease in an occipital cluster. (**C**) Left: Cluster-corrected correlation between the individual coupling strength increases from childhood to adolescence (difference adolescence – childhood) and recall performance improvement (delayed recall_adolescence – delayed recall_childhood). Asterisks denote significant electrodes. Subjects with higher coupling strength increases showed stronger recall performance improvements from childhood to adolescence. This effect was strongest at electrode F3: rho = 0.56, p=0.0008 (right, scatter plot with linear trend line). (**D**) Correlation between coupling strength increase (co-occurrence corrected) and sleep-dependent memory consolidation enhancement ([delayed recall_adolescence – immediate recall_adolescence] - [delayed recall_childhood – immediate recall_childhood]) from childhood to adolescence at electrode F3: rho = 0.54, p=0.0011. This indicates that subjects with a higher developmental increase in coupling strength show higher sleep benefits on memory consolidation.

The online version of this article includes the following figure supplement(s) for figure 4:

**Figure supplement 1.** Preferred phase development, correlations with behavior and non-individualized parameters.

correlations, which left the significant correlation unchanged (*Recall Performance*: p.rho = 0.57, p<0.001; *Sleep-dependent memory consolidation*: p.rho = 0.54, p=0.0013). Finally, we directly compared the strengths of the dependent correlations (coupling to memory vs. spindle density to memory formation) using a percentile bootstrap approach (*Wilcox, 2016*) and found that behavioral memory metrics were significantly better predicted by coupling strength than spindle density correlations (*Recall performance*: Z = 2.20, p=0.028, $CI_{95}$ = [0.048 0.99]; *Sleep-dependent memory consolidation*: Z = 2.17, p=0.030; $CI_{95}$ = [0.040 1.018]). Furthermore, all reported correlations were not driven by demographic parameters or differences in immediate recall scores as shown by partial correlations (correlation of coupling strength and recall performance with the following confounding variables: p.rho_{age difference} = 0.63, p<0.001; p.rho_{pubertal stage} = 0.58, p<0.001, p.rho_{IQ} = 0.56, p<0.001; p.rho_{immediate recall} = 0.46, p=0.009; correlation of coupling strength and sleep-dependent

memory consolidation: $p.rho_{age\ difference}$ = 0.54, p=0.001, $p.rho_{pubertal\ stage}$ = 0.54, p=0.001, $p.rho_{IQ}$ = 0.55, p=0.001, $p.rho_{immediate\ recall}$ = 0.53, p=0.002). We also confirmed that non-individualized preferred phase and coupling strength values did not predict sleep-dependent memory consolidation enhancements (*Figure 4—figure supplement 1D,E*; for how analytical choices impacted the correlations see *Figure 4—figure supplement 1G*).

## Discussion

In a longitudinal sleep and memory study, we show that SO and spindles become more precisely coupled during brain maturation from childhood to adolescence. Crucially, this increase indicated improved recall performance and sleep-dependent consolidation in a declarative memory task. Collectively, our findings suggest that the emergence of precise temporal SO-spindle coordination might index the development of hippocampal-neocortical memory systems.

Here, we employed an individualized cross-frequency coupling approach, since profound changes in the network organization during development were apparent, which potentially confound cross-frequency coupling estimates (*Aru et al., 2015*; *Helfrich et al., 2018b*): In particular, we observed (1) changes in non-oscillatory 1/f background activity (*Figure 2C*) and (2) low frequency power (*Figure 2D* and *Figure 2—figure supplement 1A,B*) as well as (3) a peak frequency shift in the spindle-band (*Figure 2E,F*). While some SO and spindle activity markers, such as amplitude and density have been previously associated with memory consolidation (*Gais et al., 2002*; *Huber et al., 2004*; *Schabus et al., 2004*; *Schabus et al., 2006*), our findings now could provide a mechanistic explanation how improved network coordination subserves memory formation. Furthermore, our results highlight the critical need to account for individual oscillatory features and to discount non-oscillatory broadband effects, which are known to confound cross-frequency coupling analyses (*Figure 4—figure supplement 1D,E,G*; *Aru et al., 2015*; *Bódizs et al., 2009*; *Cole and Voytek, 2017*; *Ujma et al., 2015*; *Voytek et al., 2015*). While our findings replicate the pattern that parietal spindles are faster than frontal spindles, we did not find reliable evidence for the presence of two distinct spindle peaks in individual electrodes (*Figure 2—figure supplement 2*).

Importantly, SO-spindle coupling is one of the main building blocks of the active system memory consolidation theory, which posits that sleep-dependent memory consolidation is coordinated by a temporal hierarchy of SO, sleep spindles and hippocampal ripples (*Clemens et al., 2011*; *Diekelmann and Born, 2010*; *Helfrich et al., 2019*; *Helfrich et al., 2018b*; *Klinzing et al., 2016*; *Klinzing et al., 2019*; *Mölle et al., 2002*; *Mölle et al., 2006*; *Muehlroth et al., 2019*; *Niethard et al., 2018*; *Rasch and Born, 2013*; *Staresina et al., 2015*). Recently, it has been shown, that only SO coupled spindles, not isolated spindles trigger hippocampal-neocortical information transfer, suggesting that SO-spindle coupling might be a proxy of this subcortical-cortical network communication that is measurable on the scalp level (*Helfrich et al., 2019*). While we did not measure hippocampal activity in the current study, our results still provide direct evidence for the behavioral relevance of the temporal hierarchy of SO and sleep spindles. We demonstrate that improved temporal coordination between prefrontal SOs and sleep spindles indexes memory network maturation. Notably, we observed that the precise coupling phase over frontal EEG sensors was already determined during childhood, but over time even more spindles became precisely locked to the preferred SO phase (~0°; SO 'up-state': Figure 3G,H & *Figure 4—figure supplement 1A*). This might indicate that SO-spindle coupling is an inherent feature of the human memory system.

Previous research has emphasized the importance of a preferred phase closer to the SO up-state (*Helfrich et al., 2018b*; *Muehlroth et al., 2019*; *Winer et al., 2019*). However, those studies focused on the detrimental impact of aging on memory consolidation. Now our findings reveal that mechanisms of maturation and aging are different. While maturation reduces the circular variance by increasing the coupling strength, (*Figure 4A,B*), aging leads to a temporal dispersion of the coupling away from the SO 'up-state' with almost constant circular variance (*Helfrich et al., 2018b*; *Muehlroth et al., 2019*). In both instances, only changes in frontal cross-frequency coupling were predictive of behavior, which is in line with the hypothesized origin of SOs in the medial prefrontal cortex (*Massimini et al., 2004*; *Murphy et al., 2009*). Critically, precise prefrontal SO-spindle coupling governs over hippocampal-neocortical network communication and actively initiates the information transfer within the network (*Helfrich et al., 2019*). Therefore, our results indicate that brain maturation leads to more fluent communication within memory networks.

A multitude of studies suggested a general positive effect of sleep on memory consolidation (*Backhaus et al., 2008*; *Diekelmann and Born, 2010*; *Huber et al., 2004*; *Klinzing et al., 2019*). Accordingly, our data provide additional support for this consideration by showing that consolidation is superior after sleep retention than after a wake period during adolescence (*Figure 1—figure supplement 1C*). In the present study, SO-spindle coupling strength did not predict sleep specific memory improvements (*Figure 4—figure supplement 1F*). Two likely explanations possibly contributed to this observation: (1) Performance in the word pair task was close to ceiling level in the adolescent group. Considering, that most evidence for a relationship between SO-spindle coupling and memory formation stems from studies comparing across relatively wide age ranges (*Helfrich et al., 2018b*; *Muehlroth et al., 2019*), an overall high performance level could lead to less variance that might mitigate detecting a relationship between coupling strength and sleep specific memory benefits. (2) Apart from memory consolidation, the coupling strength increase is also related to maturation of general cognitive abilities (*Figure 1C* and *Figure 4C*; also see *Hahn et al., 2019*), which further complicates disentangling sleep-specific memory benefits since both processes exhibit a substantial overlap (*Hoedlmoser et al., 2014*; *Schabus et al., 2004*; *Schabus et al., 2006*).

Nonetheless, several recent findings demonstrated a causal relationship between SO-spindle coupling and memory formation. For example, ripple-triggered electrical stimulation (*Maingret et al., 2016*) or optogenetic manipulation boosted SO-spindle coupling and memory performance the next day in rodents (*Latchoumane et al., 2017*). In humans, it has been shown that electrical stimulation at ~0.75 Hz induces both SO and spindle power as well as improved memory performance (*Marshall et al., 2006*). However, recent replication attempts provided contradictory findings (*Bueno-Lopez et al., 2019*; *Lafon et al., 2017*; *Sahlem et al., 2015*). Apart from electrical stimulation, slow rocking motions (*Perrault et al., 2019*) and auditory stimulation (*Ngo et al., 2013*) might be effective in entraining SOs and concomitant spindles to improve memory performance. Furthermore, targeted memory reactivation triggers SO-spindle coupling, which enabled successful decoding of mnemonic information (*Antony et al., 2018*; *Antony et al., 2019*; *Cairney et al., 2018*; *Schönauer, 2018*; *Schönauer et al., 2017*). Taken together, several lines of research converge on the notion that precise SO-spindle coupling constitutes to a key mechanism for memory formation.

This is of immediate relevance for future studies, since it has been shown that for example tau pathology in the medial temporal lobe, a precursor of Alzheimer's disease, also impairs prefrontal SO-spindle coupling (*Winer et al., 2019*). Thus, cross-frequency coupling might potentially constitute a novel pathway to understand age- or disease-related cognitive decline, amendable to intervention. In future, it might be possible to entrain SO-spindle synchrony through electrical (*Lustenberger et al., 2016*) or auditory stimulation (*Ngo et al., 2013*) to alleviate memory deficits (for a recent review see *Hanslmayr et al., 2019*).

# Materials and methods

**Key resources table**

| Reagent type (species) or resource | Designation | Source or reference | Identifiers | Additional information |
|---|---|---|---|---|
| Software, algorithm | Brain Vision Analyzer 2.2 | Brain Products GmbH https://www.brainproducts.com | RRID:SCR_002356 | |
| Software, algorithm | CircStat 2012 | *Berens, 2009* https://philippberens.wordpress.com/code/circstats/ | RRID:SCR_016651 | |
| Software, algorithm | EEGLAB 13_4_4b | *Delorme and Makeig, 2004* https://sccn.ucsd.edu/eeglab/index.php | RRID:SCR_007292 | |
| Software, algorithm | FieldTrip 20161016 | *Oostenveld et al., 2011* http://www.fieldtriptoolbox.org/ | RRID:SCR_004849 | |

*Continued on next page*

*Continued*

| Reagent type (species) or resource | Designation | Source or reference | Identifiers | Additional information |
|---|---|---|---|---|
| Software, algorithm | IRASA | *Wen and Liu, 2016* https://purr.purdue.edu/publications/1987/1 | | |
| Software, algorithm | MATLAB 2015a | MathWorks Inc | RRID:SCR_001622 | |
| Software, algorithm | Presentation software | Neurobehavioral Systems, Inc http://www.neurobs.com | RRID:SCR_002521 | |
| Software, algorithm | Somnolyzer 24 × 7 | Koninklijke Philips N.V. https://www.philips.co.in | | |

## Participants

Initially, 63 subjects (mean ± SD age, 9.56 ± 0.76 years; 28 female, 35 male) were recruited during childhood from public elementary schools (*Hoedlmoser et al., 2014*). Seven years later, 36 healthy subjects agreed to participate again in the current follow-up study. Two participants were excluded because of technical issues during polysomnography (PSG). One participant was excluded because of insufficient amount of NREM3 sleep (2.63%). All analyses are based on 33 healthy participants (23 female) during childhood (mean ± SD age, 9.5 ± 0.8 years) and during adolescence (mean ± SD age, 16 ± 0.9 years). Participants and their legal custodian provided written informed consent before entering the study. The study protocol was conducted in accordance with the Declaration of Helsinki and approved by the ethics committee of the University of Salzburg (EK-GZ:16/2014).

## Experimental design

All participants were screened for possible sleep disorders using established questionnaires at both time points (Children's Sleep Habits Questionnaire [*Owens et al., 2000*]; Pittsburgh Sleep Quality Index [*Buysse et al., 1989*]). To determine the pubertal stage of the participants we used the Pubertal Development Scale (*Carskadon and Acebo, 1993*). Cognitive abilities were assessed by the Wechsler intelligence Scale for Children (*Petermann and Petermann, 2007*) and the Wechsler Adult Intelligence Scale (*Wechsler, 1997*). Participants maintained a regular sleep rhythm during the study as verified by wrist actigraphy (Cambridge Neurotechnology Actiwatch, Cambridge, UK) and a sleep log (*Saletu et al., 1987*). To guarantee a habitual sleep environment, full-night ambulatory polysomnography was recorded at the participants' homes. Sleep was recorded during two nights during both childhood and adolescence. The first night was used for adaptation purposes. The experimental night served as retention interval for the word pair task. To satisfy the differences in sleep need during maturation, participants had a scheduled time in bed of 10 hr during childhood (8.30 pm – 6.30 am) and 8 hr during adolescence (11.00 pm – 7.00 am). The word pair task at the experimental night consisted of word pair encoding followed by an immediate recall after a short delay (10 min) in the evening and a delayed recall after a sleep retention interval in the morning (*Figure 1A*). To confirm that sleep has a beneficial effect on memory, 31 participants performed the word pair task in a counterbalanced wake condition during adolescence. In the wake condition, participants encoded new word pairs at 8.00 am in the morning and recalled them after 10 hr of wakefulness (*Figure 1—figure supplement 1C*).

## Word pair task

Participants performed a previously established declarative memory task (*Figure 1B*), where they encoded and recalled non-associated word pairs (*Gais et al., 2002*; *Hoedlmoser et al., 2014*; *Schabus et al., 2004*; *Schabus et al., 2008*). To alleviate the impact of enhanced cognitive abilities across maturation on task performance, we adjusted word pair count and timing parameters in order to increase the difficulty during adolescence. Participants had to encode 50 pairs during childhood and 80 word pairs during adolescence. The word pair task was performed using Presentation software (Version 18.2, Neurobehavioral Systems, Inc, Berkeley, CA, www.neurobs.com). During childhood, each pair was visible for 5 s followed by a fixation cross for 3 s. During adolescence, each pair was presented for 6.5 s followed by a fixation cross for 3.5 s. Participants were advised to imagine a

visual connection between the two words in order to control for different mnemonic strategies. All word pairs were presented twice in randomized order. During recall, only the first word of the word pair was presented. Participants had 10 s to recall the corresponding missing word during childhood and 6.5 s during adolescence. If the participants recalled the corresponding missing word, they had to press the mouse button and name the word. A button press or running out of recall time was followed by a fixation cross for 1.5 s during childhood and 3.5 s during adolescence as a reference interval. It was not allowed to name already disappeared word pairs. Participants received no feedback about their performance. Words were presented in randomized order in the immediate and delayed recall block.

## Sleep recording and sleep staging

Ambulatory PSG was recorded with an Alphatrace, Becker Meditec (Karlsruhe, Germany) portable amplifier system using gold-plated electrodes (Grass Technologies, AstroMed GmbH, Germany) at a sampling rate of 512 Hz. Eleven EEG-electrodes were placed on the scalp according to standard 10–20 system. Two electromyogram electrodes were placed at left and right musculus mentalis. Two horizontal electrooculogram electrodes were placed above the right outer canthus and below the left outer canthus, with two additional vertical electrooculogram electrodes above and below the right eye as well as two electrodes placed on bilateral mastoids. The EEG signal was referenced online against Cz and re-referenced offline to a common average reference. For sleep staging, electrodes were re-referenced to contra lateral mastoids. Sleep stages were automatically scored in 30 s bins (Somnolyzer 24 × 7, Koninklijke Philips N.V.; Eindhoven, The Netherlands) and visually controlled by an expert scorer according to standard sleep staging criteria (*Iber et al., 2007*).

## Word pair task data analysis

Recall performance (*Figure 2C*) was calculated as percentage by dividing the number of correctly recalled word pairs and semantically correct word pairs by the total count of word pairs. Semantically correct word pairs were weighted by 0.5. A word pair was rated as semantically correct whenever the answer was unambiguously related to the correct answer (e.g. 'boot' instead of 'shoe'). Recall performance development was subsequently calculated by subtracting delayed recall performance during childhood from the performance during adolescence.

Sleep-dependent memory consolidation was calculated by subtracting immediate recall scores from delayed recall scores. The developmental change of sleep-dependent memory consolidation was calculated by subtracting values during childhood from values during adolescence.

## EEG data preprocessing

EEG data were visually inspected using BrainVision Analyzer 2 (Brain Products GmbH, Germany). Artefactual activity was marked for every 5 s bin of continuous data for further processing in FieldTrip (*Oostenveld et al., 2011*) and EEGlab (*Delorme and Makeig, 2004*). Segments containing artifacts were rejected for all following analyses.

## Power spectra and disentangling 1/f fractal from oscillatory components

Average power spectra from 0.1 to 30 Hz were calculated by means of a Fast Fourier Transform (FFT) after applying a Hanning window on continuous 15 s NREM sleep data (i.e. NREM2 and NREM3; *Figure 2A,B*) in 1 s sliding steps. All power values are log transformed. To mitigate power differences between childhood and adolescence (*Figure 2A*) we z-normalized the continuous signal on every channel in the time domain (*Figure 2B*). To disentangle the 1/f fractal component from the true oscillatory components we applied irregular auto-spectral analysis (IRASA, *Wen and Liu, 2016*) on the normalized data from 0.1 to 30 Hz in a sliding window of 15 s in 1 s steps. In brief, the EEG signal in each window is stretched by a non-integer resampling factor (rf; e.g. 1.1) and subsequently compressed by a corresponding rf* (e.g. 0.9). Resampling was repeated with factors from 1.1 to 1.9 in 0.05 steps whereby the corresponding rf* is calculated by 2-rf. This resampling causes peak shifts of the oscillatory components in the frequency domain. The 1/f component of the signal however remains unchanged. Because resampling is done in a pair-wise fashion, median averaging across resampled FFT segments extracts the fractal 1/f component power spectrum (*Figure 2C*) by

averaging out oscillatory components. Finally, to obtain the true oscillatory power spectrum (*Figure 2D*) we subtracted the extracted fractal component power spectrum from the mixed (i.e. containing fractal and oscillatory parts) power spectrum in the semi-log space (*Figure 2B*). Based on the oscillatory power spectrum we detected slow oscillation (<2 Hz) and sleep spindle (10–18 Hz) frequency peaks (*Figure 2E,F*) with their corresponding amplitude. Note however, that the extracted amplitude was 1/f corrected. We repeated this step for every subject during both time points in both nights on every channel in order to obtain individual SO and sleep spindle frequency peaks. These values were used for individualized event detections of SOs and sleep spindles in the subject-time-night-channel domain. We also considered the possibility of two distinct spindle frequency peaks. However, most of the participants only expressed a single frequency peak in the spindle range (*Figure 2—figure supplement 2A*) even though we also observed the expected fronto-posterior frequency gradient (*Figure 2—figure supplement 2B*). Therefore, we only considered the highest peak in the power spectrum after discounting the fractal component as the most representative spindle frequency at the corresponding electrode.

### Event detection

We devised individualized event detection on every channel separately by adjusting previously established algorithms (*Helfrich et al., 2018b*; *Mölle et al., 2011*; *Staresina et al., 2015*) based on the obtained individual SO and spindle features described above.

For SO detection, the continuous EEG signal was high-pass filtered at 0.16 Hz and subsequently low-pass filtered at 2 Hz. Next, we detected all zero-crossings of the filtered signal. A zero-crossing was considered a slow oscillation if it fulfilled the time criterion (length 0.8–2 s) and exceeded the 75% percentile threshold of the amplitude. Valid slow oscillation epochs were extracted ±2.5 s around the trough of artifact free data.

For sleep spindle detection, we bandpass filtered the continuous signal ±2 Hz around the individual spindle peak frequency. Next, we extracted the instantaneous amplitude by a Hilbert transform and subsequently smoothed the signal with a 200 ms moving average. We considered a valid spindle event if the signal exceeded the 75% percentile threshold of the amplitude for 0.5 to 3 s. Sleep spindle epochs were extracted ±2.5 s around the peak of artifact free data. To alleviate the prominent power differences (*Figure 2A*) we further z-normalized all detected SO and spindle events within each subject for subsequent analyses (*Figure 3D* and *Figure 3—figure supplement 1*). Spindle density was computed as the mean spindle number per 30 s NREM3 epoch.

### Full-night time-frequency representations

For the full-night time-frequency representations (*Figure 3A,B*), we first segmented the data in 30 s epochs with an 85% overlap. Then we applied a multi-taper spectral analysis using 29 discrete prolate slepian sequences (dpss) tapers with a frequency smoothing of ±0.5 Hz from 0.5 to 30 Hz in 0.5 Hz steps (*Mitra and Pesaran, 1999*; *Prerau et al., 2017*).

### Event co-occurrence

To obtain the co-occurrence rate of detected sleep spindles and slow oscillations (*Figure 3C*), we quantified how many SO events coincided with a sleep spindle peak within a 2.5 s time interval and subsequently divided the measure by overall detected spindle events. We chose the 2.5 s time window to capture ±2 SO cycles around the spindle peak (*Helfrich et al., 2019*). Because of the low co-occurrence rate during NREM2 we conducted all following analyses on NREM3 event detections only.

### Event-locked time-frequency representations

For event-locked time-frequency representations (*Figure 3G, H*, left), we first applied a 500 ms Hanning window on normalized SO-trough locked segments (−2 to 2 s, in 50 ms steps). Frequency power was analyzed from 5 to 30 Hz in 0.5 Hz steps. Next, we baseline corrected the spectral estimates by constructing a bootstrapped baseline distribution based on the −2 to −1.5 epoch of all trials (10000 iterations). All values were subsequently z-transformed relative to the obtained means and standard deviations of the bootstrapped distribution.

## Event-locked cross-frequency coupling

To assess event-locked cross-frequency coupling (*Dvorak and Fenton, 2014*; *Helfrich et al., 2018b*; *Staresina et al., 2015*) we focused our analyses on NREM3 sleep given that we observed a higher co-occurrence of sleep spindles and slow oscillations during this sleep stage (*Figure 3C*). Based on the individualized and normalized spindle peak-locked epochs, we first low-pass filtered the time-locked data by 2 Hz and subsequently extracted the phase angle of the SO-component corresponding to the spindle amplitude peak using a Hilbert transform. To obtain the mean direction of the phase angles (preferred phase, *Figure 3E–H*) and coupling strength (i.e. phase-locking value, *Figure 4A*) of all NREM3 events, we used the CircStat Toolbox functions circ_mean and circ_r. However, there are no repeated measures statistical tests for circular data. Therefore, to allow for repeated measure testing, we calculated the absolute circular distance (circ_dist) of the individual preferred phase to the SO peak at 0° ('up-state'), where spindles are preferentially locked to *Mölle et al., 2011*; *Mölle et al., 2002*; *Staresina et al., 2015*. This transformation rendered our circular variable linear. The spindles within preferred phase metric (*Figure 4B*) was calculated by dividing the number of spindles within ±22.5° of the preferred phase (i.e. 45° radius) by the total number of spindle events. To control for differences in event number, we used bootstrapping by randomly drawing 500 spindle events 100 times and recalculating the mean direction and coupling strength (*Figure 4—figure supplement 1C*).

## Statistical analyses

We calculated two-factorial repeated measure ANOVAs to assess the differences in recall performance between childhood and adolescence (*Figure 1C*) and to investigate the co-occurrence of sleep spindle- and SO-events during NREM2 and NREM3 sleep (*Figure 3C*). We used cluster-based random permutation testing (*Maris and Oostenveld, 2007*) to correct for multiple comparison (Monte-Carlo method, cluster alpha 0.05, max size criterion, 1000 iterations, critical alpha level 0.05 two-sided). We clustered the data in the frequency (*Figure 2A–C*), space (*Figure 2E,F Figure 4A–C*, *Figure 2—figure supplement 1*, *Figure 4—figure supplement 1A,B,E*) and time domain (*Figure 3D*). For correlational analyses we calculated spearman rank correlations (*Figure 4C,D*, *Figure 4—figure supplement 1B,E,F*), circular-linear correlations (*Figure 4—figure supplement 1B,E* middle column) and circular-circular correlations (as implemented in CircStat toolbox (*Berens, 2009*; *Figure 4—figure supplement 1C*). For cluster-corrected correlations we transformed the correlation coefficients (rho) to t-values (p<0.05 threshold). Watson-Williams-Test (circular ANOVA) was not appropriate for repeated measure designs, therefore we transformed circular measures (phase) to parametric values by calculating the absolute distance to 0° (i.e. SO 'up-state') to allow for repeated measure testing (*Figure 4—figure supplement 1A*, topographical plot). Partial eta squared ($\eta^2$), Cohen's d (d) and spearman correlation coefficients (rho) are reported for effect sizes. We estimated cluster effect size by calculating the effect size for every data point within the significant cluster (in frequency, space and time) separately, followed by averaging across all obtained effect sizes. To compare correlations, we used a percentile bootstrap method for overlapping correlations (*Wilcox, 2016*). In brief we randomly drew participants with replacement, while keeping the dependency between the observation pairs of the correlations. Next we calculated two spearman correlation coefficients and their difference. This step was repeated 1000 times. Based on the resulting distribution we computed the 95% confidence interval ($CI_{95}$) of the difference and p-value.

## Data analyses

Data were analyzed with MatLab 2015a (Mathworks Inc), utilizing functions from the Fieldtrip toolbox (*Oostenveld et al., 2011*), EEGlab toolbox (*Delorme and Makeig, 2004*) and CircStat toolbox (*Berens, 2009*) as well as custom written code. For filtering we used the EEGlab function eegfilt.m and the FieldTrip function ft_preprocessing.m. For time domain analyses we used ft_timelockanalysis.m. Frequency and time-frequency domain analyses were conducted with the ft_freqanalysis.m function. For irregular auto-spectral analysis (IRASA *Wen and Liu, 2016*) we used code published in the original research paper. Cluster-based permutation tests were carried out with the ft_freqstatistics.m function. Circular statistics were computed with the CircStat functions circ_mean.m (preferred phase), circ_sd.m (circular standard deviation of the preferred phase), circ_r.m (plv, coupling strength), circ_dist.m (circular distance), circ_corrcc.m (circular-circular correlation)

and circ_corrcl.m (circular-linear correlation). Results were plotted using circ_plot.m (circular plots, CircStat), topoplot.m (topographical plots; EEGlab) and MatLab functions (plot.m, imagesc.m).

## Acknowledgements

This research was supported by Austrian Science Fund (T397- B02, P25000), the Jacobs Foundation (JS1112H) and the Centre for Cognitive Neuroscience Salzburg (CCNS). MAH was additionally supported by the Doctoral Collage 'Imaging the Mind' (FWF, Austrian Science Fund W1233-G17). RFH is supported by the Hertie Foundation (Hertie Network of Excellence in Clinical Neuroscience), the Baden-Württemberg Foundation and the German Research Foundation (DFG, HE 8329/2-1).

The authors are grateful to Judith Roell and Ann-Kathrin Joechner for their assistance during the data acquisition. We also thank all of the children who participated in this study and their parents as well as the principals of the schools and the local education authority (Mag. Dipl. Paed. Birgit Heinrich, Prof. Mag. Josef Thurner) in Salzburg who supported this study.

## Additional information

### Funding

| Funder | Grant reference number | Author |
|---|---|---|
| Austrian Science Fund | W1233-G17 | Michael A Hahn |
| Austrian Science Fund | T397-B02, P25000 | Kerstin Hoedlmoser |
| Hertie Foundation | Hertie Network for Excellence in Clinical Neuroscience | Randolph F Helfrich |
| Baden-Württemberg Foundation | Postdoctoral fellowship | Randolph F Helfrich |
| Jacobs Foundation | JS1112H | Kerstin Hoedlmoser |
| Centre for Cognitive Neuroscience Salzburg (CCNS) | | Kerstin Hoedlmoser Michael A Hahn |
| German Research Foundation | HE 8329/2-1 | Randolph F Helfrich |

The funders had no role in study design, data collection and interpretation, or the decision to submit the work for publication.

### Author contributions

Michael A Hahn, Conceptualization, Data curation, Software, Formal analysis, Validation, Investigation, Visualization, Writing - original draft; Dominik Heib, Data curation, Writing - review and editing; Manuel Schabus, Resources, Methodology, Writing - review and editing; Kerstin Hoedlmoser, Conceptualization, Resources, Data curation, Supervision, Funding acquisition, Validation, Investigation, Methodology, Project administration, Writing - review and editing; Randolph F Helfrich, Conceptualization, Data curation, Software, Formal analysis, Supervision, Validation, Visualization, Methodology, Writing - review and editing

### Author ORCIDs

Michael A Hahn ⓘ https://orcid.org/0000-0002-3022-0552
Manuel Schabus ⓘ http://orcid.org/0000-0001-5899-8772
Kerstin Hoedlmoser ⓘ https://orcid.org/0000-0001-5177-4389
Randolph F Helfrich ⓘ https://orcid.org/0000-0001-8045-3111

### Ethics

Human subjects: The study protocol was conducted in accordance with the Declaration of Helsinki and approved by the ethics committee of the University of Salzburg (EK-435 GZ:16/2014). Participants and their legal custodian provided written informed consent before entering the study.

Decision letter and Author response
Decision letter https://doi.org/10.7554/eLife.53730.sa1
Author response https://doi.org/10.7554/eLife.53730.sa2

## Additional files

### Supplementary files

- Transparent reporting form

### Data availability

Data required to reproduce the main conclusions and all figures are available at Dryad under doi https://doi.org/10.5061/dryad.8sf7m0chn.

The following dataset was generated:

| Author(s) | Year | Dataset title | Dataset URL | Database and Identifier |
|---|---|---|---|---|
| Hahn MA, Heib DPJ, Schabus M, Hoedlmoser K, Helfrich RF | 2019 | Slow oscillation-spindle coupling predicts enhanced memory formation from childhood to adolescence | http://dx.doi.org/10.5061/dryad.8sf7m0chn | Dryad Digital Repository, 10.5061/dryad.8sf7m0chn |

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
