## [Decision Letter]

**Acceptance summary:**

In this longitudinal study, Hahn et al. investigate the role of sleep oscillations in developmental changes in memory performance. They find that increased locking between slow oscillations and sleep spindles with maturation correlates with increased memory performance. This finding is interesting as it reflects both morphological and cognitive changes in the brain.

**Decision letter after peer review:**

Thank you for submitting your article "Slow oscillation-spindle coupling predicts enhanced memory formation from childhood to adolescence" for consideration by *eLife*. Your article has been reviewed by two peer reviewers, and the evaluation has been overseen by Saskia Haegens as the Reviewing Editor and Laura Colgin as the Senior Editor. The reviewers have opted to remain anonymous.

The reviewers have discussed the reviews with one another and the Reviewing Editor has drafted this decision to help you prepare a revised submission.

Summary:

In this longitudinal study, Hahn et al. investigated the role of sleep oscillations in developmental changes in memory performance. To this end they assessed memory performance using a word-pair task in one sample of kids during childhood (~9.5 yrs) and adolescence (~16 yrs). Sleep polysomnography and EEG were measured. The results show that the locking of slow oscillations and spindles increased from childhood to adolescence. This increase in locking was correlated with increased memory performance. Overall, the reviewers considered this paper of broad interest and strong methodological approach.

Essential revisions:

1) The authors assert in several places that the used word-pair task is hippocampus dependent. However, they do not cite any study that supports this statement. Indeed, one could contest this assertion by citing studies showing that word-pairs may well be associated without (or with only little) hippocampal involvement through a process called 'unitization' (see e.g. Quamme et al., 2007, Hippocampus). As this point is not really critical to the manuscript since the results are important regardless of whether the task is dependent on hippocampus or not, we suggest the authors either drop the reference to the hippocampus, or cite studies in support of this statement.

2) Related to the previous point: not all hippocampal dependent memory tests are affected by sleep, and presumably the authors base their assumption that sleep has a beneficial effect on the selected paradigm on the experiment described on the previous paper (Hahn et al., 2019) where they demonstrated superiority of memory retention following a night of sleep relative to a wake period. Still, the fact that there is no deterioration of memory retention between immediate and delayed tests for the adolescent group, and that the performance of some subjects is very close to ceiling, leads to the suspicion that this paradigm is not ideal for revealing memory changes during sleep for the older group. For the adolescent group, does the SO-spindle coupling during sleep predict the performance superiority of sleep night relative to the wake condition? This can help validate that SO-spindle coupling are related to enhanced performance in this task following sleep.

3) It would be helpful if the authors add some more info regarding the sleep-dependent performance changes between the groups? For example, what is the relationship between sleep-dependent memory consolidation in childhood to adolescence? (i.e. ([delayed recall_adolescence – immediate recall_adolescence] vs. [delayed recall_childhood – immediate recall_childhood])).

4) While Figure 4C is interesting, the conclusions drawn from the comparison of immediate-immediate and delayed-delayed tests (each test has a different number of words and the performance of the older group close to ceiling) is related to maturation of general cognitive abilities, which are discussed in the previous article, and not directly related to memory processes during sleep, so the sleep-related conclusions from this comparison should be toned down.

5) The main statement (and title) of the paper comes down to Figure 4D where the coupling strength is linked to sleep-dependent memory consolidation enhancement. How many electrodes showed a cluster-corrected correlation of this effect? Are they the same 2 electrodes as in Figure 4C or only F3? For this panel, the authors use a sub-group of spindle events that co-occur with detected SO-events – presumably this sub-group is described in Figure 4B, it's a bit difficult to understand from the text. Can the authors explain whether/how the two main differences between the groups affect the coupling strength? The main differences that may bias this index are: (1) Increased spindle amplitude (Figure 2E) between groups and (2) the different percentage of spindles co-occurring with SO. As there are several options to quantify cross-frequency coupling, it is important to understand what the measure used in this paper is sensitive to.

---

## [Author Response]

Essential revisions:1) The authors assert in several places that the used word-pair task is hippocampus dependent. However, they do not cite any study that supports this statement. Indeed, one could contest this assertion by citing studies showing that word-pairs may well be associated without (or with only little) hippocampal involvement through a process called 'unitization' (see e.g. Quamme et al., 2007, Hippocampus). As this point is not really critical to the manuscript since the results are important regardless of whether the task is dependent on hippocampus or not, we suggest the authors either drop the reference to the hippocampus, or cite studies in support of this statement.

We thank the reviewers for their remark and apologize for the confusion. We fully agree that it is still unclear to what extent the hippocampus is involved during word-pair tasks. We now describe the learning task as a declarative word pair task instead of a hippocampus-dependent task. We changed the manuscript in several instances:

Abstract:

“Here, we use a longitudinal study design spanning from childhood to adolescence, where participants underwent polysomnography and performed a word-pair learning task.”

Results:

“At both time points participants underwent full-night ambulatory polysomnography at their home during two nights (adaptation and experimental night; Figure 1A) and performed a declarative memory task during the experimental night (Figure 1B).”

Figure 1A legend:

“(a) Longitudinal study design. […] At the following experimental night, participants performed a declarative word pair learning task during which they encoded and recalled semantically non-associated word pairs before sleep.”

Materials and methods:

“Word pair task

Participants performed a previously established declarative memory task (Figure 1B), where they encoded and recalled non-associated word pairs (Hoedlmoser et al., 2014; Schabus et al., 2004; Schabus et al., 2008).”

2) Related to the previous point: not all hippocampal dependent memory tests are affected by sleep, and presumably the authors base their assumption that sleep has a beneficial effect on the selected paradigm on the experiment described on the previous paper (Hahn et al., 2019) where they demonstrated superiority of memory retention following a night of sleep relative to a wake period. Still, the fact that there is no deterioration of memory retention between immediate and delayed tests for the adolescent group, and that the performance of some subjects is very close to ceiling, leads to the suspicion that this paradigm is not ideal for revealing memory changes during sleep for the older group. For the adolescent group, does the SO-spindle coupling during sleep predict the performance superiority of sleep night relative to the wake condition? This can help validate that SO-spindle coupling are related to enhanced performance in this task following sleep.

We thank the reviewers for this remark regarding memory tasks and sleep. In the following, we divide our answers addressing the concerns about the sleep effect on the word pair task we used in our study.

Beneficial effect of sleep on the memory paradigm:

As the reviewers correctly noted, we have previously shown that sleep constitutes to memory consolidation (Hahn et al., 2019). We apologize for not providing the necessary evidence to substantiate our claim that sleep has a beneficial effect on word pair memory (Backhaus et al., 2008; Gais et al., 2002; Hahn et al., 2019; Schabus et al., 2004) in the present study. Therefore, we conducted additional analyses by comparing a sleep retention interval with a wake retention interval during adolescence. We found that memory consolidation (delayed – immediate recall) is superior in the sleep than in the wake condition (t(30) = 6.04, p <.001, d = 1.08, Figure 1—figure supplement 1C), demonstrating that sleep helps maintaining memory, whereas wakefulness leads to forgetting.

We now report the results of the sleep vs. wake condition analysis in the new Figure 1—figure supplement 1.

We also now report and discuss this analysis in the main text and added additional information about the wake condition to the Materials and methods section:

Results:

“During adolescence, memory consolidation was superior after a sleep retention interval compared to a wake retention interval (Figure 1—figure supplement 1C), indicating a beneficial effect of sleep on memory.”

Discussion:

“A multitude of studies speaks for the general positive effect of sleep on memory consolidation (Backhaus et al., 2008; Diekelmann and Born, 2010; Huber et al., 2004; Klinzing et al., 2019). Accordingly, our data provides additional support for this consideration by showing that consolidation is superior after sleep retention than after a wake period during adolescence (Figure 1—figure supplement 1C).”

Experimental design:

“To confirm sleep has a beneficial effect on memory, 31 participants performed the word pair task in a counterbalanced wake condition during adolescence. In the wake condition, participants encoded new word pairs at 8.00 am in the morning and recalled them after 10 h of wakefulness (Figure 1—figure supplement 1C).”

Revealing memory changes during sleep in the adolescent group:

Especially with declarative tasks, the beneficial sleep effect is not necessarily translated in an overnight memory improvement, but rather a maintenance of memory performance after sleep. In other words, sleep attenuates forgetting compared to wakefulness. This assertion is further supported by the comparison of a sleep and a wake condition (Figure 1—figure supplement 1C). Actual sleep related memory improvements in declarative tasks (e.g. Backhaus et al., 2008) can be partially explained by providing additional feedback during the learning phase or after the first recall. However, in the current study we did not provide feedback to our participants. This could be a possible explanation for the overnight memory maintenance and not overnight memory improvement in the word pair task in our current study (see Figure 1C). Further, as the reviewers correctly observed, the performance of some subjects is very close to ceiling levels during adolescence, which might have prevented overnight memory improvements during adolescence. However, with the additional analysis of the wake condition now provided in Figure 1—figure supplement 1, we can demonstrate that sleep has an effect on the memory paradigm during adolescence.

We now discuss the issues of ceiling level performance and the sleep effect on memory during adolescence and their relationship to SO-spindle coupling in the manuscript:

“A multitude of studies suggested a general positive effect of sleep on memory consolidation (Backhaus et al., 2008; Diekelmann and Born, 2010; Huber et al., 2004; Klinzing et al., 2019). […] Considering, that most evidence for a relationship between SO-spindle coupling and memory formation stems from studies comparing across relatively wide age ranges (Helfrich et al., 2018b; Muehlroth et al., 2019), an overall high performance level could lead to less variance that might mitigate detecting a relationship between coupling strength and sleep specific memory benefits.”

Coupling strength as a predictor for sleep specific performance enhancements:

Following the reviewers’ suggestion, we correlated the coupling strength during adolescence with the difference in memory consolidation between the sleep and the wake condition ([delayed recall_sleep_ – immediate recall_sleep]_ – [delayed recall_wake_ –immediate recall_wake_]). Coupling strength was not significantly correlated with the sleep specific memory benefit (rho = -0.05, p = 0.787, see Figure 4—figure supplement 1F).

We also now refer to this new analysis in the Results section:

“To validate whether the SO-spindle coupling strength predicts sleep specific memory formation, we correlated coupling strength during adolescence with the difference in memory consolidation between the sleep and wake condition ([(delayed recall_sleep_ –immediate recall_sleep_) – (delayed recall_wake_ –immediate recall_wake)_]). However, we could not find evidence for this relationship (Figure 4—figure supplement 1F).”

As the reviewers correctly observed, the memory performance during adolescence approached ceiling level, an issue that leads to overall less variance in the sample. This might be a constituting factor to why coupling strength was not predictive of the sleep specific memory. Indeed, the most recent evidence for the behavioral relevance of SO-spindle coupling stems from samples comparing young adults to older adults (Helfrich et al., 2018b; Muehlroth et al., 2019), thus having more variance in the memory performance and coupling metrics. Following this notion, we initially analyzed to which account the *changes* of SO-spindle coupling predict *changes* of memory retention and sleep-dependent memory consolidation. With this analytic consideration, we could show that maturation of SO-spindle coupling is predictive of memory network maturation (Figure 4C, D).

The results of these additional analyses (Figure 4—figure supplement 1F) are now further examined in the Discussion section of the manuscript:

“A multitude of studies suggested a general positive effect of sleep on memory consolidation (Backhaus et al., 2008; Diekelmann and Born, 2010; Huber et al., 2004; Klinzing et al., 2019). […] Considering, that most evidence for a relationship between SO-spindle coupling and memory formation stems from studies comparing across relatively wide age ranges (Helfrich et al., 2018b; Muehlroth et al., 2019), an overall high performance level could lead to less variance that might mitigate detecting a relationship between coupling strength and sleep specific memory benefits.”

3) It would be helpful if the authors add some more info regarding the sleep-dependent performance changes between the groups? For example, what is the relationship between sleep-dependent memory consolidation in childhood to adolescence? (i.e. ([delayed recall_adolescence – immediate recall_adolescence] vs. [delayed recall_childhood – immediate recall_childhood])).

We thank the reviewers for their suggestion. We agree that additional information about the relationship of memory consolidation between the two maturational stages is helpful to interpret our results.

To better describe the sleep-dependent performance during both time points, we directly compared the difference of delayed – immediate recall during childhood and adolescence. We found no significant difference between the memory consolidation after a sleep retention interval between childhood and adolescence (t(32) = -1.43, p = 0.161, d = -0.35). This additional analysis reflects the interaction effect between maturation and recall time (F_1,32_ = 2.059, p = 0.161, η^2^ = 0.06) we report in the Results section. Note, that the coupling strength increase still predicts the increase in sleep-dependent memory consolidation on a single-subject level across maturation (rho = 0.54, p = 0.0011, Figure 4D). In addition, there was no correlation between the sleep-dependent performance changes during childhood and adolescence (rho = -0.01, p = 0.965; I.e. [delayed recall_adolescence_ – immediate recall_adolescence_] with [delayed recall_childhood_ – immediate recall_childhood_ ]).

These additional analyses are now reported in Figure 1—figure supplement 1.

These results are also now referenced in the Results section:

“As previously shown (Hahn et al., 2019), memory recall improved from childhood to adolescence (Figure 1C; F_1,32_ = 38.071, p < 0.001, η^2^ = 0.54) and immediate recall was better than delayed recall (F_1,32_ = 6.408, p = 0.016, η^2^ = 0.17; Maturation*Recall Time interaction: F_1,32_ = 2.059, p = 0.161, η^2^ = 0.06). Next, we assessed the relationship of sleep-dependent memory consolidation (delayed recall – immediate recall) between childhood and adolescence and found no correlation between the two maturational stages (Figure 1—figure supplement 1A; for a direct comparison of sleep-dependent memory consolidation see Figure 1—figure supplement 1B).”

4) While Figure 4C is interesting, the conclusions drawn from the comparison of immediate-immediate and delayed-delayed tests (each test has a different number of words and the performance of the older group close to ceiling) is related to maturation of general cognitive abilities, which are discussed in the previous article, and not directly related to memory processes during sleep, so the sleep-related conclusions from this comparison should be toned down.

We agree with the reviewers’ concern. We did not intend to overstate our findings. Accordingly, we changed the wording throughout the manuscript to be more appropriate to our results. Specifically, we now state in the manuscript:

Abstract:

“After disentangling oscillatory components from 1/f activity, we found frequency shifts within SO and spindle frequency bands. […] Critically, this increase indicated enhanced memory formation from childhood to adolescence.”

Discussion summary:

“In a longitudinal sleep and memory study, we show that SOs and spindles become more precisely coupled during brain maturation from childhood to adolescence. […] Collectively, our findings suggest that the emergence of precise temporal SO-spindle coordination might index the development of hippocampal-neocortical memory systems.”

On the mechanistic relevance of SO-spindle coupling for memory consolidation:

“While some SO and spindle activity markers, such as amplitude and density have been previously associated with memory consolidation, our findings now could provide a mechanistic explanation how improved network coordination subserves memory formation.”

On SO-spindle coupling being a marker for the human memory system:

“We demonstrate that improved temporal coordination between prefrontal SOs and sleep spindles indexes network maturation. […] This might indicate that SO-spindle coupling is an inherent feature of the human memory system.”

Additionally we now discuss Figure 4C in the light of the additional behavioral analyses conducted for Figure 1—figure supplement 1:

“In the present study, SO-spindle coupling strength did not predict sleep specific memory improvements (Figure 4—figure supplement 1F). […] (2) Apart from memory consolidation, the coupling strength increase is also related to maturation of general cognitive abilities (Figure 1C and Figure 4C; also see Hahn et al., 2019), which further complicates disentangling sleep-specific memory benefits since both processes exhibit a substantial overlap (Hoedlmoser et al., 2014; Schabus et al., 2004; Schabus et al., 2006).”

5) The main statement (and title) of the paper comes down to Figure 4D where the coupling strength is linked to sleep-dependent memory consolidation enhancement. How many electrodes showed a cluster-corrected correlation of this effect? Are they the same 2 electrodes as in Figure 4C or only F3? For this panel, the authors use a sub-group of spindle events that co-occur with detected SO-events – presumably this sub-group is described in Figure 4B, it's a bit difficult to understand from the text. Can the authors explain whether/how the two main differences between the groups affect the coupling strength? The main differences that may bias this index are: (1) Increased spindle amplitude (Figure 2E) between groups and (2) the different percentage of spindles co-occurring with SO. As there are several options to quantify cross-frequency coupling, it is important to understand what the measure used in this paper is sensitive to.

We thank the reviewers for making us aware that we did not explain the rationale and usage of the different coupling parameters detailed enough. We fully agree that this issue makes it difficult to understand from the text which metric was used and how it influences the other coupling metrics. The reviewers raise several important issues, which we now have indexed for point by point responses.

First, we would like to reiterate our cross-frequency coupling analyses strategy. While cross-frequency coupling can be a value tool to assess brain network communication is bears several pitfalls that could confound the robustness of the analyses. Several recent publications (Aru et al., 2015; Cole and Voytek, 2017; Dvorak and Fenton, 2014; Gerber et al., 2016; Kramer et al., 2008; Scheffer-Teixeira and Tort, 2016) indicate that cross-frequency coupling can be severely impacted by (1) shifts in 1/f dynamics, (2) the absence of oscillations in the amplitude providing high frequency signal and the phase providing low frequency signal and (3) power differences between the compared groups (as power influences the precision of phase estimation).

Our analyses strategy encompassed several steps to mitigate these pitfalls. (1) We controlled for changes in 1/f dynamics by subtracting the fractal component from the power spectra (Figure 2C, D). (2) This step was further necessary to first, prove the presence of oscillations by showing clearly discernable SO and spindle peaks (Figure 2D and Figure 2—figure supplement 2A) and second, to inform the spindle detection algorithm using the individualized peak data to ensure the detection of valid spindle oscillation events.

We further mitigated the problem of the absence of oscillations by only assessing the coupling strength for spindle detections that were in close temporal proximity (co-occurred) with SO detections (Figure 3C and Figure 4—figure supplement 1D, E, G).

How the absence of co-occurring events influences the coupling analyses is illustrated in Figure 4—figure supplement 1D, where we computed our analyses across both sleep stages (NREM2 and NREM3) which entails a high number of isolated spindle events (without the presence of a SO) in NREM2 (Figure 3C).

3) To overcome the clearly apparent power difference between childhood and adolescence (Figure 2A) we z-normalized all spindle events before extracting the corresponding SO phase. As indicated by Figure 3D there was no significant difference between childhood and adolescence in the phase providing SO-filtered signal and spindle event around the respective spindle peak at 0 s. Thus, making it unlikely that a power difference would have impacted our results.

In the following, we will respond to the specific issues raised by the reviewers point by point:

How many electrodes showed a cluster-corrected correlation in Figure 4D:

The topographical distribution of the correlation coefficients for the correlation between coupling strength increase and sleep-dependent memory consolidation enhancement (Figure 4D) is depicted in Author response image 1.

**Author response image 1. sa2fig1:** (A) Topographical distribution of the correlations between maturational coupling strength increase and enhanced sleep-dependent memory consolidation (related to Figure 4D). (B) Topographical distribution of the correlations between maturational coupling strength increase and recall performance increase.

The distribution of the effect was highly similar to the correlation analyses between coupling strength increase and recall performance improvement (Author response image 1B and Figure 4C). Therefore, we only showed the scatter plot for electrode F3. According to the reviewers’ suggestion we also computed cluster-corrected correlation analyses and found that only F3 showed a cluster-corrected correlation between couplings strength increase and sleep-dependent memory consolidation enhancement (rho = 0.54, p = 0.023, cluster-corrected). Notably, it is important to consider that cluster size is also dependent on the arbitrarily chosen cluster-alpha, which only creates a threshold for considering a sample as a candidate member for the cluster (Maris and Oostenveld, 2007; Oostenveld et al., 2011). Further, because of the rather low spatial coverage of the ambulatory EEG, the detection of large clusters is also less likely. Taken together our results show, that co-occurrence corrected coupling strength increases predict sleep-dependent memory consolidation enhancements on a single subject level after cluster-correction

Subgroup of spindles in preferred phase depicted in Figure 4B:

We agree with the reviewers that we did not describe the rationale of the analyses for Figure 4B clearly enough and combined with the other (interrelated) coupling parameters makes it hard for the reader to follow the analyses. The subgroup of spindles that is depicted in Figure 4B differs from the subgroup of spindles in Figure 4D. Figure 4B serves the purpose to illustrate the coupling strength increase from childhood to adolescence, which in other words translates to more spindles arriving in the preferred SO-phase. In Figure 4D we are showing the coupling strength that is corrected for co-occurrence (i.e. coupling strength computed only for events that co-occur with a detected SO-event). To improve comprehensibility, we added explanatory sentences to the manuscript about the results in Figure 4B and 4D:

Figure 4B:

“To further illustrate the impact of the coupling strength increase on spindle-SO-events, we calculated the percentage of spindle events that peaked at the preferred phase (± 22.5°) as compared to all detected events (Figure 4B, left). […] These results reveal an overall increase in SO-spindle coupling precision from childhood to adolescence.”

Figure 4D:

“Next, we only considered spindle events that co-occur with detected SO-events to (1) ensure more precise coupling metrics by guaranteeing the presence of oscillations and (2) to follow the notion that the temporal proximity of these events is crucial for sleep-dependent memory consolidation (Diekelmann and Born, 2010; Helfrich et al., 2018b; Muehlroth et al., 2019; Rasch and Born, 2013). Participants with a stronger coupling strength increase also showed enhanced sleep-dependent memory consolidation (delayed recall – immediate recall) from childhood to adolescence (rho = 0.54, p = 0.0011; Figure 4D, electrode F3).”

Figure 4D legend:

“(D) Correlation between coupling strength increase (co-occurrence corrected) and sleep-dependent memory consolidation enhancement ([delayed recall_adolescence_ – immediate recall_adolescence]_ – [delayed recall_childhood_ – immediate recall_childhood_]) from childhood to adolescence at electrode F3: rho = 0.54, p = 0.0011. This indicates that subjects with a higher developmental increase in coupling strength show higher sleep benefits on memory consolidation.”

Influence of group difference on the coupling strength:

Concerning SO spindle co-occurrence rate (Figure 3C): In order to compute reliable coupling estimates of SO and sleep spindles it is paramount to ensure that both events that are presumably coupled are actually present at the same time in the signal. Otherwise, this could cause spurious coupling estimates (see Figure 4—figure supplement 1D, E, G). Therefore, we conducted the co-occurrence analysis in order to assess, whether it is actually reasonable to compute SO-spindle coupling parameters in NREM2. SOs can occur in sleep stage NREM2; however, they are not the defining feature of this sleep stage. When comparing both developmental stages (childhood, adolescence) and sleep stages (NREM2, NREM3) using a repeated measure ANOVA, the co-occurrence rate was overall higher in NREM3 than in NREM2 (F(1.00, 32.00) = 2334.191, p < 0.001). However, we found no overall difference between the developmental stages (F(1.00, 32.00) = 0.001, p = 0.971) and only a trend for a interaction between sleep stage and developmental stage (F(1.00, 32.00) = 3.973, p = 0.055). Because of the overall low co-occurrence rate in NREM2, we decided to restrict our analyses to NREM3 sleep to avoid spurious coupling estimates. Comparing co-occurrence rate between childhood and adolescence during NREM3 sleep showed no significant effect (t(32) = -0.9108, p = 0.37). Further, for computing the coupling strength, we were only considering spindle events that co-occurred with SOs based on the ±2.5s time criterion and thus we are already controlling for differences in co-occurrence rate.

Nonetheless, we partialed out the difference in co-occurence rate in NREM3 from the correlations between behavior and coupling strength and found the correlations unchanged:

Recall Performance: p.rho = 0.55, p = 0.0010

Sleep-dependent memory consolidation: p.rho = 0.53, p = 0.0017

We also added a description of what the co-occurrence rate reflects in the Results section of the manuscript to prevent confusion with the percentage of spindles in preferred phase metric:

“Next, we quantified how many separate SO and spindle event detections co-occurred within a 2.5 s time window (reflecting ± 2 SO cycles around the spindle peak (Helfrich et al., 2019)). […] Co-occurrence rate was higher in NREM3 than NREM2 sleep during childhood and adolescence (Figure 3C; F_1,32_ = 2334.19, p < 0.001, η^2^ = 0.99). Subsequently we restricted our analyses to NREM3 sleep to avoid spurious cross-frequency coupling estimates caused by the lack of simultaneous detections during NREM2 sleep (for circular plots including NREM2 sleep see Figure 4—figure supplement 1D).”

Concerning the sleep spindle amplitude differences between childhood and adolescence and whether that could have affected the coupling analyses: The reviewers are correct that differences in power/amplitude can be a major confound in cross-frequency coupling analyses (Aru et al., 2015; Cole and Voytek, 2017; Gerber et al., 2016; Kramer et al., 2008; Scheffer-Teixeira and Tort, 2016). In order to mitigate this problem, we combined several methodological considerations to ensure viable coupling estimates. First, for detections of sleep spindles and SOs we were utilizing relative amplitude criteria (75-percentile threshold) to allow for comparable detection rates. Second, all detected events were subsequently z-normalized to alleviate amplitude differences (see Figure 3D). Note that we found no amplitude differences in SO-filtered spindle event around the peak. This is crucial because systematic differences in the phase-providing signal (SO-component of the spindle locked trial) can systemically affect the accuracy of the phase estimate. Therefore, we are confident that our coupling analyses are not confounded by amplitude differences in *detected* spindle events.

Considering the mentioned significant spindle amplitude differences that are depicted in Figure 2E, we first would like to note that the amplitude differences are clustered on posterior electrodes and not on frontal leads where we identified the relevant behavioral effects.

Critically, these spindle amplitude differences were also not related to any behavioral changes (see Figure 2—figure supplement 1C).

However, to rule out that the spindle amplitude differences could bias the coupling strength index we also computed partial correlations between the behavioral measures and coupling strength while controlling for spindle amplitude differences and found the correlations unchanged:

Recall Performance: p.rho = 0.57, p < 0.001

Sleep-dependent memory consolidation: p.rho = 0.54, p = 0.0013

To clarify the relationship between spindle amplitude and cross-frequency coupling estimates, we edited the manuscript in two ways. Specifically we now state:

To ensure reliable coupling estimates, we further Z-normalized individual spindle-locked data epochs in the time domain (Figure 3D) for all subsequent analyses to avoid possible confounding amplitude differences (Aru et al., 2015; Cole and Voytek, 2017; Helfrich et al., 2018b). Differences in the grand average spindle time-lock directly reflect the enhanced SO-spindle coupling, which becomes visible in the time domain when more events are precisely coupled to the SO ‘up-state’. This effect can also be appreciated in single subject spindle-locked data (Figure 3E, F, left). Note that we found no differences in the underlying SO-component around the spindle peak (0 s, time point of phase readout), thus, confirming that the Z-normalization alleviated possible amplitude differences (Figure 3D, inset; for non-normalized spindle-locked data see Figure 3—figure supplement 1).

We also edited the legend of Figure 3D:“(D) Grand average of z-normalized sleep spindle events (mean ± SEM) during childhood (blue) and adolescence (red) at electrode Fz with the corresponding SO-low-pass filtered (< 2Hz) EEG-trace (inset). […] Further note, no amplitude differences in the SO-filtered signal around the spindle peak at 0 s (i.e. time point of the phase readout).”

Different approaches to assess cross-frequency coupling and how they relate to the current event-locked coupling approach:

We thank the reviewer for this remark, which indeed is an essential consideration when studying cross-frequency interactions. First, it is important to highlight that all metrics are derived from the same time series through filtering and applying a Hilbert transform to extract the phase series of the low frequency component and the amplitude series of the faster frequency. The key question then becomes how the interaction between these two is quantified. Multiple methods have been introduced in the past (PLV: e.g. (Mormann et al., 2005); Modulation Index (Canolty et al., 2006); Kullback-Leibler Divergence Modulation Index (Tort et al., 2008); Oscillatory-triggered coupling (Dvorak and Fenton, 2014). For a recent comparison see also Hulsemann et al. 2019.

Since this is a major concern, we addressed this issue in great detail in previous publications.

First, a direct comparison of the employed coupling metric (which is conceptually similar to the oscillatory triggered coupling metric) with the classic Canolty MI indicated that both metrics are highly correlated (see Helfrich et al., 2018b, rho = 0.7645, p < 0.0001). In the same vein, we conducted a more direct comparison between these coupling metrics in a different study recently (Helfrich et al., 2018a, see Supplementary Figure 4), and found high significant correlation between all compared metrics (Helfrich et al., 2018a, see Supplementary Figure 4).

Second, given that multiple artifacts can give rise to spurious cross-frequency coupling (Aru et al., 2015; Cole and Voytek, 2017; Gerber et al., 2016; Scheffer-Teixeira and Tort, 2016), we adopted a conservative analytical approach, which only quantifies coupling after the presence of clear oscillations has been established.

These issues are discussed in the Results section:

“Co-occurrence rate was higher in NREM3 than NREM2 sleep during childhood and adolescence (Figure 3C; F_1,32_ = 2334.19, p < 0.001, η^2^ = 0.99). […] To ensure reliable coupling estimates, we further Z-normalized individual spindle-locked data epochs in the time domain (Figure 3D) for all subsequent analyses to avoid possible confounding amplitude differences (Aru et al., 2015; Cole and Voytek, 2017; Helfrich et al., 2018b).”

To address the relationship between the coupling metric in the present study and other cross-frequency coupling methods, as well as how they can be impacted by differences in the underlying signal, we added two statements to the manuscript. We now specifically state:

“To quantify the interplay of SO and spindle oscillations, we employed event-locked cross-frequency analyses (Dvorak and Fenton, 2014; Helfrich et al., 2018b; Staresina et al., 2015). […] Therefore, we adopted a conservative approach by first alleviating power differences (Figure 2B and Figure 3D) and establishing the presence of oscillations in the signal (Figure 2D and Figure 3C).”

And

“Next, we only considered spindle events that co-occur with detected SO-events to (1) ensure more precise coupling metrics by guaranteeing the presence of oscillations and (2) to follow the notion that the temporal proximity of these events is crucial for sleep-dependent memory consolidation (Diekelmann and Born, 2010; Helfrich et al., 2018b; Muehlroth et al., 2019; Rasch and Born, 2013). Participants with a stronger coupling strength increase also showed enhanced sleep-dependent memory consolidation (delayed recall – immediate recall) from childhood to adolescence (rho = 0.54, p = 0.0011; Figure 4D, electrode F3).”